# Inflorescence Transcriptome Sequencing and Development of New EST-SSR Markers in Common Buckwheat (*Fagopyrum esculentum*)

**DOI:** 10.3390/plants11060742

**Published:** 2022-03-10

**Authors:** Yang Liu, Xiaomei Fang, Tian Tang, Yudong Wang, Yinhuan Wu, Jinyu Luo, Haotian Wu, Yingqian Wang, Jian Zhang, Renwu Ruan, Meiliang Zhou, Kaixuan Zhang, Zelin Yi

**Affiliations:** 1College of Agronomy and Biotechnology, Southwest University, Chongqing 400716, China; 15236393773@163.com (Y.L.); swufxm@swu.edu.cn (X.F.); ttian19980721@126.com (T.T.); yudongwangswu@outlook.com (Y.W.); wyh994aaa@126.com (Y.W.); tenderisrain@gmail.com (J.L.); wuhaotian12138@163.com (H.W.); w15730095008@163.com (Y.W.); zhangjianswau@126.com (J.Z.); rrwryysm@163.com (R.R.); 2Institute of Crop Sciences, Chinese Academy of Agricultural Sciences, Beijing 100081, China; zhoumeiliang@caas.cn

**Keywords:** transcriptome, RNA-seq, EST-SSR, DEGs

## Abstract

Common buckwheat (*Fagopyrum esculentum* M.) is known for its adaptability, good nutrition, and medicinal and health care value. However, genetic studies of buckwheat have been hindered by limited genomic resources and genetic markers. In this study, Illumina HiSeq 4000 high-throughput sequencing technology was used to sequence the transcriptome of green-flower common buckwheat (Gr) with coarse pedicels and white-flower Ukrainian daliqiao (UD) with fine pedicels. A total of 118,448 unigenes were obtained, with an average length of 1248 bp and an N50 of 1850 bp. A total of 39,432 differentially expressed genes (DEGs) were identified, and the DEGs of the porphyrins and chlorophyll metabolic pathway had significantly upregulated expression in Gr. Then, a total of 17,579 sequences containing SSR loci were detected, and 20,756 EST-SSR loci were found. The distribution frequency of EST-SSR in the transcriptome was 17.52%, and the average distribution density was 8.21 kb. A total of 224 pairs of primers were randomly selected for synthesis; 35 varieties of common buckwheat and 13 varieties of Tartary buckwheat were verified through these primers. The clustering results well verified the previous conclusion that common buckwheat and Tartary buckwheat had a distant genetic relationship. The EST-SSR markers identified and developed in this study will be helpful to enrich the transcriptome information and marker-assisted selection breeding of buckwheat.

## 1. Introduction

Common buckwheat (*Fagopyrum esculentum* Moench) is a medicinal and edible crop that belongs to the eudicot family *Polygonaceae* [1]. Due to its characteristics of a shorter growing period, wide adaptability to different geographical environments, and strong resistance to extreme climates, common buckwheat is widely cultivated in temperate regions of Asia, Europe, and North America [2]. Common buckwheat seeds are rich in protein, fat, starch, vitamin, rutin, mineral elements, and vegetable cellulose, and it has preventive and therapeutic effects on cardiovascular diseases, diabetes, and constipation [3]. Therefore, a large number of studies were mainly focused on the biologically active ingredients of buckwheat, such as flavonoids and flavones [4], phytosterols [5], and fagopyrins [6].

Although common buckwheat has high nutritional value, the seed yield is low because of its self-incompatibility. The lack of genomic resources and tightly linked markers of important agronomic genes of buckwheat is an important factor restricting the molecular breeding of common buckwheat [7]. Molecular markers, which take DNA as the carrier of genetic information and the polymorphism of DNA between individuals and the linkage between its related traits as genetic markers, have been widely used in crop genetic improvement and variety selection [8]. So far, many types of molecular markers have been developed and used for genetic diversity and relationship studies, linkage maps construction and quantitative trait locus (QTL) mapping in common buckwheat, such as RAPD (random amplified polymorphic DNA) [9,10], AFLP (amplified fragment length polymorphism) [11,12], SNP (single-nucleotide polymorphism) [13,14], and SSR (simple sequence repeat) [15,16,17,18]. 

SSR markers have been widely used in the analysis of plant genetic diversity, construction of genetic maps, gene mapping, and cloning because of their locus specificity, co-dominant inheritance, multi-allelic nature, high polymorphism, and are well distributed throughout the genome [19,20]. Based on the original sequences for the development of SSR markers, SSRs can be divided into two categories: genomic SSRs, which are identified from random genomic sequences, and expressed sequence tag (EST) SSRs, which are identified from transcribed RNA sequences. The traditional development of genomic SSRs is inefficient and costly [21,22] since it requires the construction of partial genomic DNA libraries and labor-intensive Sanger sequencing [23]. In contrast, EST-SSRs can be rapidly mined at a lower cost from expressed sequences, which are more conserved and have a higher level of transferability between related species [24,25,26]. Next-generation sequencing (NGS), especially de novo transcriptome sequencing, was found to be a simple and effective method with much higher throughput and relatively low cost; it also provides fascinating opportunities to develop a large number of EST-SSR markers and dramatically improves the efficiency of molecular marker development in common buckwheat, such as detection of the candidate gene (*S-ELF3*) controlling the short-styled phenotype of common buckwheat [27], differentially expressed genes in inflorescence transcriptomes between common and Tartary buckwheat [28] and genetic diversity [15,16,17,29]. However, compared with the widely used SSR primers for other crops, only a few SSR markers have been developed and reported for genetic studies in common buckwheat.

In this study, we used the Illumina sequencing platform to conduct the inflorescence transcriptome sequencing analysis of two common buckwheat varieties, namely, green-flower buckwheat (Gr) and Ukrainian daliqiao (UD). Large-scale transcriptome sequences were assembled and annotated, and a set of EST-SSRs was developed. Then, we employed the newly developed SSR markers to verify the genetic relationships and diversity among different varieties of common buckwheat and Tartary buckwheat.

## 2. Results

### 2.1. RNA Sequencing and Functional Annotation of Unigenes 

To establish the transcriptome library, two replicates of the total RNA of Gr_F and UD_F were extracted and sequenced on the Illumina HiSeq platform. The raw reads were uploaded in the NCBI Sequence Read Archive (SRA) under the accession numbers SRR17325563, SRR17325562, SRR17325561, and SRR17325560. In this study, there were 48,327,536 raw reads for Gr_F_1 and 42,364,332 raw reads for Gr_F_2, while there were 54,440,478 raw reads for UD_F_1 and 51,885,264 raw reads for UD_F_2. After removing adaptors and low-quality data, Gr_F_1, Gr_F_2, UD_F_1, and UD_F_2 produced 47,630,570, 41,788,726, 52,870,344, and 50,344,286 clean reads, respectively. The GC content reached 45.61%, 45.58%, 45.52%, and 45.54%, respectively, and the Q20 values were all greater than 97% (Appendix A). 

According to the clean reads, Trinity software generated 177,125 transcripts with an average length of 932 bp and an N50 of 1657 bp, the minimum length was 201 bp, and the maximum length was 16,788 bp. The longest cluster obtained via Corset hierarchical clustering was identified as a unigene, where 118,448 unigenes were obtained after the calculations with an average length of 1248 bp and an N50 of 1850 bp; the minimum and maximum values were the same as the former but the median length value was 882 bp. As for the length interval, the largest number of unigenes were between 501 and 1000 bp in length with 34,910 and the fewest were less than 301 bp in length with 9402 (Table 1).

Among the 118,448 assembled unigenes, seven databases were compared to obtain comprehensive gene function information. The results showed that 67,950 (57.36%) of the unigenes had significant similarity in the NCBI non-redundant (Nr) database, 43,056 (36.35%) of them were in the NT database, and 57,262 (48.34%) were in the Swiss-Prot database. In total, there were 10,798 (9.11%) unigenes annotated to all seven databases and 77,428 (65.36%) unigenes annotated in at least one database successfully (Appendix A). 

After GO annotation of the unigenes, the unigenes that were successfully annotated were classified according to the next level of three GO categories of “Biological process”, “Cellular component”, and “Molecular function” (Appendix A). According to the statistical results, the unigenes could be classified into 24 terms in “Biological process”, and the top three largest categories were “cellular process”, “metabolic process”, and “single-organism process.” There were 21 terms in “Cellular component”, and “cell”, “cell part”, and “organelle” were highly represented. In terms of “Molecular function”, there were 10 terms, of which, “binding” and “catalytic activity” were the first and second most abundant categories.

A total of 21,928 (18.51%) unigenes were annotated in the KOG database and clustered into 26 functional groups. Among the groups, the top three cluster groups were O—“Posttranslational modification, protein turnover, chaperones” (2975, 13.57%), R—“General function prediction only” (2893, 13.19%), and J—“Translation, ribosomal structure and biogenesis” (1831, 8.33%). The smallest clustered groups were X—“Unnamed protein” (1, 0.004%), N—“Cell motility” (22, 0.10%), and W—“Extracellular structures” (91, 0.41%); meanwhile, 1361 (6.20%) unigenes were annotated in group S—“Function unknown” (Appendix A).

KEGG is a database resource for understanding high-level functions and utilities of the biological system; we used KOBAS [30] software to test the statistical enrichment of differential expression unigenes in KEGG pathways. In this study, out of all the 118,448 unigenes, 28,120 (23.74%) were significantly matched to the KEGG pathway database and assigned to five biochemical pathways (hierarchy 1), including 19 main pathways (hierarchy 2) (Appendix A). In these five main classes, metabolism had the largest proportion (11,722, 55.14%), genetic information processing followed closely at (5876, 27%), while the last three were environmental information processing (1163, 5.47%), cellular processes (1391, 6.54%), and organismal systems (1106, 5.21%). The top three in the biochemical pathways (hierarchy 2) were carbohydrate metabolism; translation; and folding, sorting, and degradation.

Comparison of the assembled transcripts with the proteome of *Beta vulgris*, *Vitis vinifera*, *Theobroma cacao*, *Jatropha curcas*, and *Nelumbo nucifera* showed that 67,950 of the 118,448 unigenes were able to show similarity to at least one plant. More than 50% of these assembled transcripts had more than 60% sequence similarity, 26.5% of transcripts had 80% to 95% sequence similarity, and even 1.5% of transcripts had more than 95% sequence similarity between species (Figure 1A). According to the data, the species with the greatest similarity to common buckwheat was *Beta vulgaris* (24.6%) [31], followed by *Vitis vinifera* (12.9%) [32], *Theobroma cacao* (3.8%) [33], *Jatropha curcas* (3.4%), *Nelumbo nucifera* (3.4%), and others (51.8%) (Figure 1B).

### 2.2. Differential Expression Analysis

The results of observing the pedicels of Gr and UD at the full flowering stage showed that the pedicel diameter of Gr was thicker and showed an extremely significant difference. The contents of chlorophyll A, chlorophyll B, and total chlorophyll in the inflorescence in Gr were significantly higher than in UD (Figure 2).

To detect differentially expressed genes (DEGs), the expression levels of these unigenes in the Gr_F and UD_F were estimated. Using differential expression analysis of genes in the inflorescence of them, 19,484 genes and 25,393 genes were specifically expressed among them. A total of 39,432 unigenes were differentially expressed between the two cultivars, of which, 23,100 genes were significantly upregulated and 16,332 genes were significantly downregulated in the inflorescence of Gr_F (Appendix A). 

In the porphyrins and chlorophyll metabolic pathway [34,35,36], chlorophyll a and chlorophyll b, in addition to catalytic synthesis pathways in which the original chlorophyll acid divinyl fat are converted back into the original chlorophyll acid fat of 8-ethylene reductase (DVR), did not display differentially expressed genes, and a total of 33 differentially expressed genes were detected, where 27 of them in green-flower buckwheat had significantly upregulated expression and 6 had significantly downregulated expression (Figure 3).

Flavonoids are the main nutrient in buckwheat and phenylalanine is the direct precursor of flavonoid biosynthesis. The pigments of white flowers are mainly colorless flavonoids, such as flavone and flavonols. Therefore, we analyzed the differentially expressed genes in the phenylpropane biosynthesis pathway (Ko0940) and detected 186 differentially expressed genes. Among them, there were 15 differentially expressed genes of PAL, C4H, and 4CL, 11 of which were downregulated in Gr_F, while PAL and C4H were all downregulated in Gr_F, which might have been caused by the synthesis of a large amount of flavonoids in white-flower buckwheat. The downstream genes F3H, F3’5’ H, and FLS were downregulated in Gr_F and upregulated in UD_F (Figure 4).

### 2.3. The Frequency and Distribution of SSRs

All 118,448 assembled unigenes were identified for potential SSR loci using MISA software, and a total of 20,756 SSRs were mined from 17,579 unigenes (Appendix A). The distribution frequency of EST-SSR in the transcriptome was 17.52%, and the average distribution density was 8.21 kb (Appendix A). Among the unigenes containing SSRs, 2579 unigenes had more than one SSR; meanwhile, 920 SSRs presented a compound formation (Table 2).

In the 20,756 EST-SSRs, there were 12,447 (59.97%) mono-nucleotides where the number of repetitions was 10 or more. There were 8309 motifs with two or more different base repeats, among which, trimers were the most common (4417, 53.15%), followed by dimers (3691, 44.43%), tetramers (153, 1.84%), pentamers (28, 0.34%), and hexamers (20, 0.24%) (Figure 5A).

The repeat number of most SSRs ranged from five to ten, and the most frequent repeat number was five (2861, 34.44%), followed by six (2733, 32.9%), seven (1328, 15.99%), eight (529, 6.37%), nine (342, 4.11%), ten (338, 4.07%), and more than ten (176, 2.1%) (Table 3).

Among the mono-nucleotide types, the most frequent was A/T (12,313, 59.32%), and the least frequent was C/G (134, 0.65%). Among the dimer repeats, the most frequent was AT/AT (2093, 10.08%), followed by AG/CT (1187, 5.72%), AC/GT (401, 1.93%), and CG/CG (10, 0.05%). AAG/CTT (1263, 6.09%) was the most frequent trinucleotide, and the least frequent was ACT/AGT (82, 0.9%). In addition, tetranucleotides, pentanucleotides, and hexanucleotides had fewer repeat motifs, with a total number of 201 (0.97%) (Figure 5B).

### 2.4. Primer Design and Validation of EST-SSR Markers

In this study, a total of 13,909 pairs of primers were developed from 20,756 identified EST-SSR loci, where the length of the primers was 18–23 bp, and the size of the amplified products was 100–300 bp (Appendix A). After comparing the transcriptome data of other varieties of *Fagopyrum esculentum*, part of the *Fagopyrum esculentum* genome database and the *F. tataricum* genome database, 224 pairs of SSR loci were randomly selected and synthesized (Appendix A) for homology cluster analysis of 35 varieties of common buckwheat and 13 varieties of Tartary buckwheat (Appendix A). It was found that 92 (41.07%) pairs showed polymorphism in different varieties of common buckwheat and *Tartary* buckwheat (Appendix A). The results showed that the genetic similarity coefficient of the 48 buckwheat varieties ranged from 0.38 to 0.99, and the buckwheat varieties were divided into two groups with a limit of 0.68 (Figure 6).

The first group was different varieties of common buckwheat and the second group was different varieties of Tartary buckwheat. This well verified the previous conclusion that common buckwheat and Tartary buckwheat had a distant genetic relationship and demonstrated that the SSR primers developed in this study had strong accuracy and practicability.

## 3. Discussion

In a previous study, Logacheva revealed differentially expressed genes related to sugar biosynthesis and metabolism through comparative analysis of flower and inflorescence transcriptomes of common and Tartary buckwheat [28]. High-throughput mRNA sequencing technologies were used in the genetic research of buckwheat [17]. In this study, inflorescence transcriptomes of “green-flower buckwheat” and “white-flower buckwheat” were sequenced on an Illumina HiSeq 4000 platform. Compared with the transcriptome sequencing data of immature buckwheat seeds by Shi [7], we produced a larger number of transcripts (177,125 vs. 54,975), a longer average transcript length (932 vs. 840) (Table 1), and the higher N50 value of 1657 bp indicated that we generated a high-quality assembly of the inflorescence transcriptome for common buckwheat. The assembled transcripts in this study are appropriate for transcriptome analysis, gene identification, and marker development, and could be an important source for shattering-resistant research on buckwheat in the future.

In this study, 10,798 (9.11%) unigenes were annotated using Nr, NT, Pfam, KOG, Swiss-Prot, KEGG, and GO databases, where 77,428 (65.36%) unigenes were annotated in at least one database successfully. Compared with other species, the top match was *Beta vulgaris* (24.6% sequence identity), followed by *Vitis vinifera* (12.9%), *Theobroma cacao* (3.8%), *Jatropha curcas* (3.4%), and *Nelumbo nucifera* (3.4%) (Figure 1B). The results of Shi found that the top-hits taxonomic distribution of BLAST hits of common buckwheat was from *Vitis vinifera* [7], which is consistent with our results. A similar taxonomic distribution of species also appeared in previous studies, such as buckwheat flower [28], immature seeds of buckwheat [7], and *Prunus persica* [37] Among the species closest to buckwheat, genome sequencing of *Vitis vinifera* has been completed [38], which plays an important role in genome alignment and gene annotation of buckwheat.

Common buckwheat normally has white flowers, but scientists have also bred green- and red-flower buckwheat. Breeders in Ukraine found that green buckwheat is more fertile and has larger grains [39]. Studies showed that the green-flower phenotype and stout peduncle are regulated by a recessive gene [40]. Suzuki [41] found that the green-flower buckwheat had stouter peduncles and the shattering seed ratio was lower, and Fang [42] found the petals of green-flower buckwheat contain more chlorophyll than those of white-flower buckwheat and red-flower buckwheat. Based on the previous studies, transcriptome sequencing of inflorescences of Gr and UD was conducted in this study. A total of 33 DEGs were detected in the chlorophyll synthesis pathway, of which, 27 genes were significantly upregulated in Gr; therefore, it was speculated that the green-flower buckwheat was mainly caused by chlorophyll. The results of this study laid a foundation for future research on the key candidate genes of buckwheat petal color and shattering resistance and the research on shattering resistance varieties.

A total of 20,756 SSRs were mined in 17,579 unigenes, which provided rich information for the development of the SSR marker in buckwheat. Excluding mono-nucleotide repeats, tri-nucleotide repeat motifs of EST-SSRs were the most abundant type (53.15%) of microsatellites in the study (Table 2). The results were consistent with previous studies, such as *Cucurbita pepo* [43], *Pinus tabuliformis* [44], and castor bean [45], while other results showed that di-nucleotide was the most abundant type, such as in sesame [46] and oil palm [47]. The most abundant di- and tri-nucleotide motifs in this study were AT/AT (10.08%) and AAG/CTT (6.09%) (Table 3), respectively, where the di-nucleotide was consistent with the result of Hou [48] and the tri-nucleotide was consistent with the result of Shi [7]. These results are consistent with those of most plant species previously studied [49,50]. However, our nucleotide motif frequency was slightly different from that of legumes [51] and cereals [52]. The main reasons for the difference in the distribution frequency of SSR motif types were the different SSR search criteria, the different search algorithms, and the different selection pressures between plants.

Validation of SSRs discovered via transcriptome sequencing is the next step to building a working marker set for genetic improvement efforts. In previous studies, 10 polymorphic SSR markers were utilized in genetic diversity analysis of a common buckwheat population consisting of 41 accessions of diverse origin [17], including 17 (25%) SSRs exhibited polymorphisms between *D. officinale* individuals [50]. A total of 20,756 EST-SSR loci were obtained and 13,909 pairs of primers were developed in this study, where 224 pairs of primers were synthesized and 92 (41.07%) pairs of them showed polymorphism in different varieties. The polymorphic ratio (41.07%) of the primer pairs was higher than Konishi [11] and Ma [17]. The similarity coefficient of 48 buckwheat varieties ranged from 0.38 to 0.99, and 0.68 was used as the limit to divide buckwheat varieties into two groups; most of the cultivars were grouped according to geographic distribution, mainly into Yunnan, Sichuan, Guizhou, and other regions, which indicated that the large number of new SSR markers developed in this study will be useful resources for genetic diversity analysis, genetic mapping studies, and play an important role in molecular marker-assisted selection breeding for *Fagopyrum* species.

## 4. Materials and Methods

### 4.1. Plant Materials and RNA Isolation

Two common cultivated varieties, namely, green-flower buckwheat (Gr) with green flowers and resistance to shattering and Ukraine daliqiao (UD) with white flowers and non-resistance to shattering, were cultivated in the test field at Southwest University, Chongqing, China, with normal field management during the growth period. In the full-bloom stage, the inflorescence was collected from the Gr and UD and placed in liquid nitrogen for RNA isolation. The total RNA of the samples was extracted with TRIzol Reagent (TIANGEN, Beijing, China), according to the manufacturer’s instructions. The purity and contamination of the isolated RNA were monitored on 1% agarose gels, and these RNA samples were used for cDNA library construction.

### 4.2. cDNA Library Construction and Sequence Assembly

The same amount (1.5 μg) of total RNA was taken from each sample as input for RNA sample preparations for library construction. Sequencing libraries were generated via a NEBNext^®^ Ultra™ RNA Library Prep Kit for Illumina^®^ (NEB, San Diego, CA, USA) following the manufacturer’s recommendations. In short, mRNAs were purified from total RNA using poly-T oligo-attached magnetic beads. The mRNAs were disrupted into short fragments by added fragmentation buffer. First-strand cDNA was synthesized using random hexamer primer and M-MuLV Reverse Transcriptase (RNase H), and second-strand cDNA synthesis was subsequently performed using DNA Polymerase I and RNase H. The remaining overhangs were converted into blunt ends using exonuclease/polymerase activities. The cDNA fragments with lengths of 150–200 bp were selected after purification, size selection, and adaptor ligation of the library fragments. Then, PCR was performed and PCR products were purified to establish the final cDNA libraries. The library quality was assessed on an Agilent Bioanalyzer 2100 system.

The cDNA library was sequenced on the Illumina Hiseq platform by Bioinformatics Technology Co. Ltd., Beijing, China. Raw data (raw reads) in the FASTQ format were first processed through in-house Perl scripts. Clean reads were obtained by removing reads containing adapter, ploy-N, and low-quality reads (the proportion of low-quality bases of Q-value ≤ 20 is more than 50% in a read) from raw data (raw reads). Transcriptome assembly was based on Trinity [53] with min_kmer_cov set to 2 by default, where Trinity connects the contigs and obtains sequences defined as unigenes.

### 4.3. Function Annotation and Expression Analysis of Unigenes

All of the assembled unigenes were searched in the following databases for the functional annotation and classification of unigenes: Nr (NCBI non-redundant protein sequences), Nt (NCBI non-redundant nucleotide sequences), Pfam (protein family); KOG/COG (Clusters of Orthologous Groups of proteins), Swiss-Prot (a manually annotated and reviewed protein sequence database), KO (KEGG Ortholog database), and GO (Gene Ontology). To further annotate the unigenes, GO annotation of unigenes was obtained using the Blast2GO program [54] with the cutoff E-value of 1 × 10^−6^ The unigene sequences were also aligned with the COG database to predict and classify function and the pathway annotations of these unigenes were obtained with the KEGG database [55].

Gene expression levels were estimated using RSEM [56] for each sample. The transcriptome obtained via Trinity was used as the reference sequence (ref), and the clean reads of each sample were mapped on the ref. The read count for each gene was obtained from the mapping results. Differential expression analysis of two samples was performed using the DEGseq R package [57]. The resulting *p*-values were adjusted using Benjamini and Hochberg’s method for controlling the false discovery rate, and the differentially expressed genes were defined as adjusted *p*-value < 0.05. Then, GO function enrichment analysis and KEGG metabolic pathway enrichment analysis were performed for the differentially expressed (upregulated and downregulated) genes.

### 4.4. SSR Mining and Primer Design

The Microsatellite software (MISA) [58] was used to detect the microsatellites within the unigenes in this study that were longer than 1000 bp, and the analysis parameters of MISA were set to the default. The standard of SSRs was considered to contain one to six repeat motifs in size. The minimum number of repeats of each corresponding unit size was as follows: mono—10, dimer—6, trimer—5, tetramer—5, pentamer—5, and hexamer—5. The primer pairs were designed by Primer3. The major parameters for designing SSR primers were: (1) primer length from 18 to 22 bases, (2) PCR product size ranges from 100 to 300 bp, (3) melting temperature between 55 and 61 °C with 59 °C being the optimal annealing temperature, and (4) GC content of 45–65% with an optimum of 50%. A total of 224 EST-SSR primers were randomly selected and synthesized by Beijing Genomics Institute Co. Ltd. (Beijing, China) to evaluate the application value of this set of EST-SSR markers.

### 4.5. PCR Amplification and Experimental Evaluation of Microsatellite Markers

To assess amplification efficiency and experimental evaluation of the newly developed SSRs, 35 common buckwheat accessions and 13 Tartary buckwheat accessions were used and the total genomic DNA of each accession was extracted according to a modified CTAB method [59]. The PCR amplification system consisted of 10 μL, including 1.0 μL template DNA, 1.0 μL 10X PCR buffer, 0.2 μL dNTP (2.5 mM), 1.5 μL MgCl_2_ (1.5 mM), the forward and reverse primer (1 μM each) were each 0.5 μL, and 0.1 μL Taq polymerase (2 U/μL). Finally, ddH_2_O was added to complete the 10 μL. The specific reaction time and temperature of PCR amplification were set as follows: first, denaturing at 95 °C for 5 min, then cycled 40 times of 95 °C for 30 s, 55–60 °C for 30 s, 72 °C for 30 s, and finally extending at 72 °C for 15 min. PCR amplification results were detected using polyacrylamide gel electrophoresis, and the primer bands with polymorphism were counted and processed. Coefficients of genetic similarity of the 48 kinds of buckwheat germplasm resources were calculated using the SIMQUAL program of the NTSYS-pc software [60] and a clustering graph for the materials was constructed using the UPGMA algorithm of the SAHN module.

## 5. Conclusions

In this study, transcriptome sequencing was carried out by extracting RNA from two materials: Gr_F and UD_F, where 118,448 unigenes were obtained with a total sequence length of 147,868,721 bp, and 77,424 unigenes were annotated in at least one database. A total of 20,756 EST-SSR loci were mined in 17,595 unigenes, where 13,909 pairs of primers were developed. After preliminary screening, 224 pairs of primers were randomly selected and synthesized for homology cluster analysis of 35 varieties of common buckwheat and 13 varieties of Tartary buckwheat, where the 48 buckwheat varieties were divided into two groups according to the similarity coefficient of 0.68. The results of transcriptome sequencing and assembly, primer sequencing, and differential expression analysis will provide a theoretical basis for species classification, germplasm conservation, genetic diversity analysis and molecular marker-assisted breeding of buckwheat.

## Figures and Tables

**Figure 1 plants-11-00742-f001:**
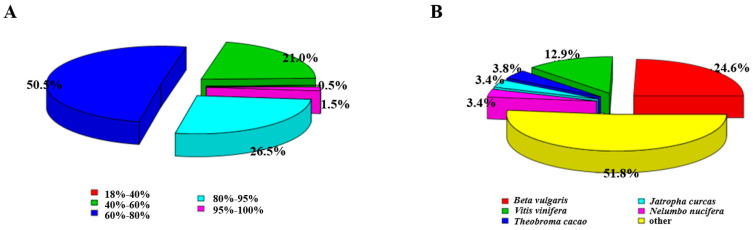
The distribution of species similarity of Nr annotations. (**A**) The similarity distribution of the Nr database. (**B**) The species classification of Nr annotations.

**Figure 2 plants-11-00742-f002:**
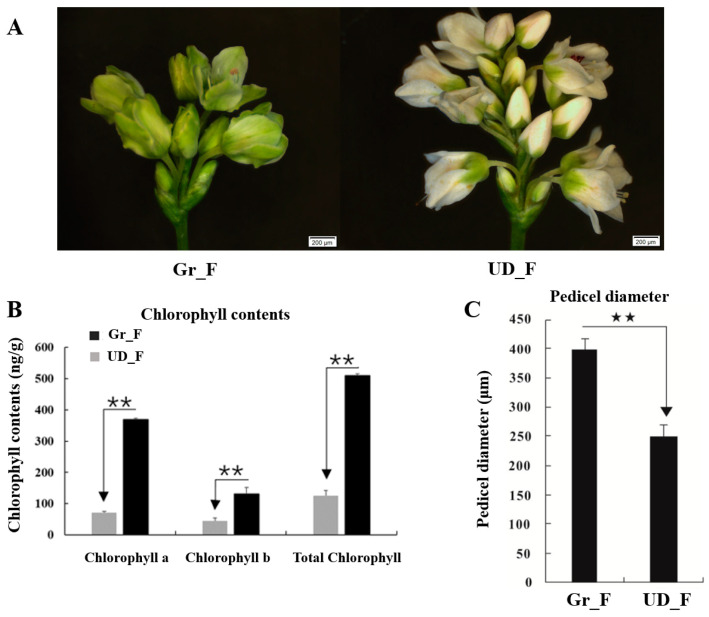
Analysis of the traits of green-flower buckwheat (Gr_F) and white-flower buckwheat (UD_F). (**A**) Electron microscopic photograph of the inflorescence. (**B**) The chlorophyll content of the flowers. (**C**) Pedicel diameter. Asterisks denote significant difference of Gr_F and UD_F (** *p* < 0.01).

**Figure 3 plants-11-00742-f003:**
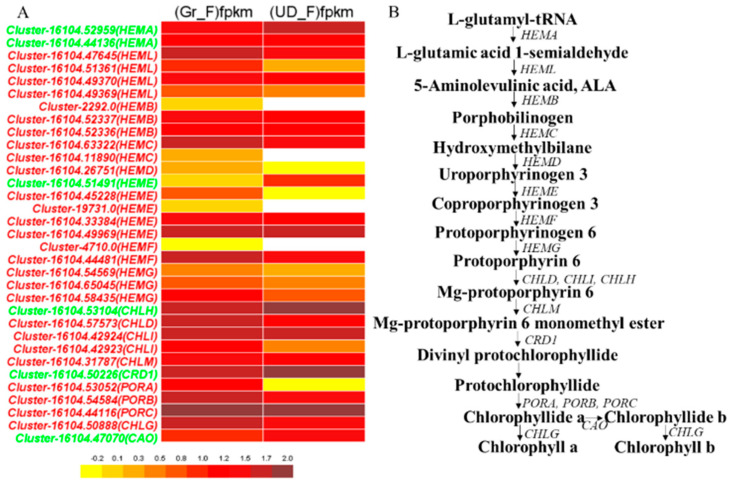
Differentially expressed genes in chlorophyll metabolism. (**A**) The DEGs in chlorophyll metabolism, where the genes ID in red show the significantly upregulated expression and the genes in green show the downregulated expression in green-flower common buckwheat. (**B**) The chlorophyll synthesis pathway. The arrows show the direction of metabolism.

**Figure 4 plants-11-00742-f004:**
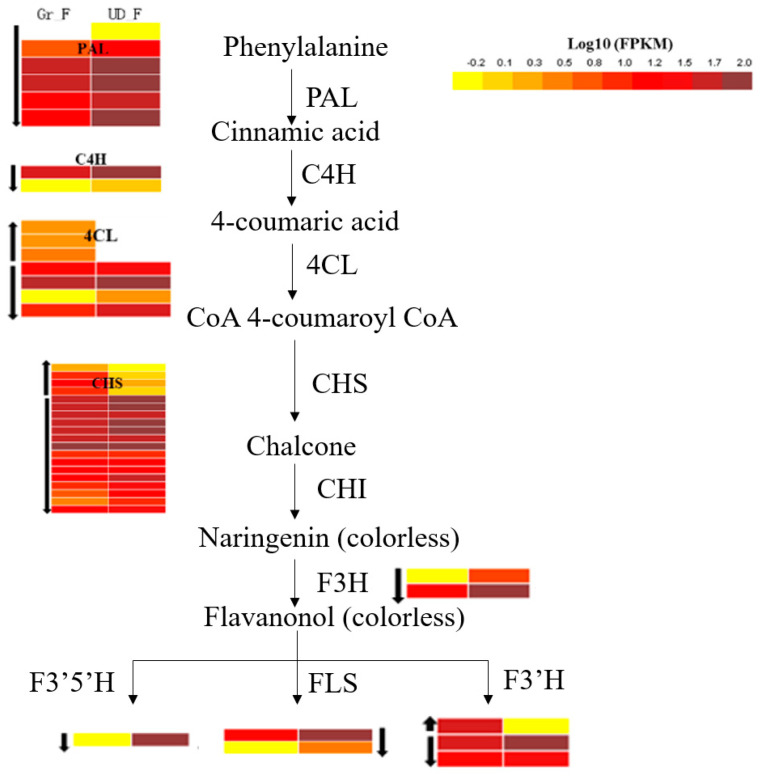
Differential expression analysis of flavonoid synthesis pathway. The color represents the amount of expression, the up and down arrows next to the expression represent upregulated or downregulated expression, respectively.

**Figure 5 plants-11-00742-f005:**
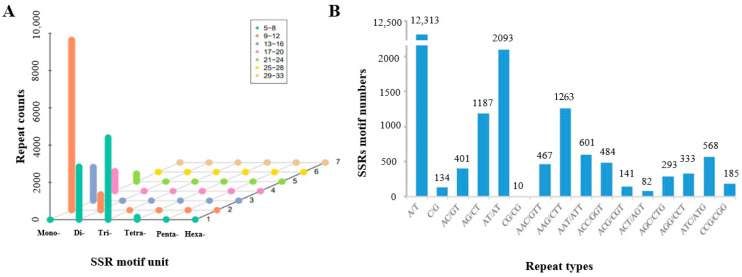
Frequency of different SSR repeat types in the entire sample. (**A**) Frequency of different motif lengths in SSRs. The *X*-axis shows the repeated SSR motif unit, the *Y*-axis shows the repeat counts, and the color shows the number of repetitions. (**B**) Frequency of EST-SSR motif types. The *X*-axis shows the SSR motif types and the *Y*-axis shows the SSR motif numbers.

**Figure 6 plants-11-00742-f006:**
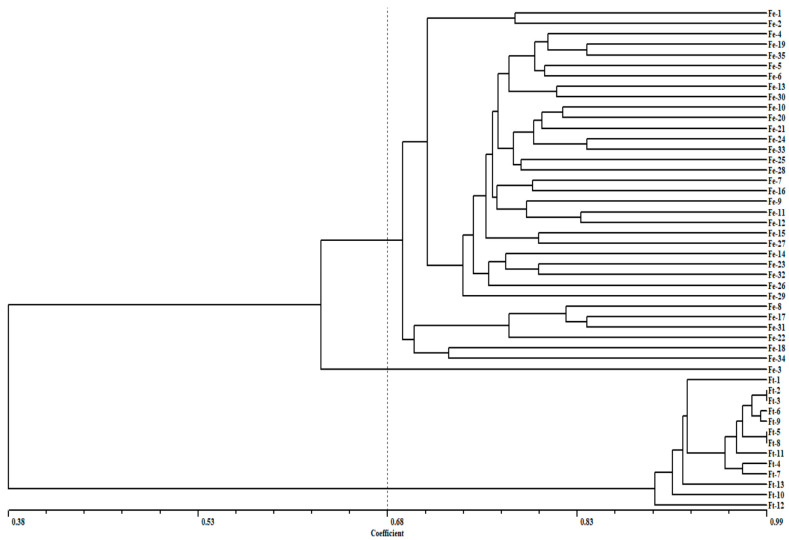
UPGMA clustering graph for buckwheat materials. The *X*-axis shows the genetic similarity coefficient.

**Table 1 plants-11-00742-t001:** Summary of the transcriptome.

Nucleotide Length (bp)	Transcripts	Genes
Min Length (bp)	201	201
Mean Length (bp)	932	1248
Median Length (bp)	514	882
Max Length (bp)	16,788	16,788
N50 (bp)	1657	1850
N90 (bp)	356	578
Total Nucleotides (bp)	165,113,675	147,868,721
Length Interval	Numbers	
<301 bp	48,071	9402
301–500 bp	38,807	20,658
501–1000 bp	36,940	34,910
1001–2000 bp	31,069	31,040
>2000 bp	22,238	22,238

**Table 2 plants-11-00742-t002:** Summary of the identified EST-SSRs.

Summary of Identified EST-SSRs	Numbers
Total number of sequences examined	118,448
Total size of examined sequence (bp)	147,868,721
Total number of identified SSRs	20,756
Number of SSR containing sequences	17,579
Number of sequences containing more than 1 SSR	2579
Number of SSRs present in compound formation	920
Mono-nucleotide	12,447
Di-nucleotide	3691
Tri-nucleotide	4417
Tetra-nucleotide	153
Penta-nucleotide	28
Hexa-nucleotide	20

**Table 3 plants-11-00742-t003:** Frequency of EST-SSR motif types. “–”: Unit Size Number of undetected duplicates.

Repeat Numbers									
Repeated Motif	5	6	7	8	9	10	>10	Total	%
A/T	-	-	-	-	-	5369	6944	12,313	59.32
C/G	-	-	-	-	-	31	103	134	0.65
AC/GT	-	181	82	54	33	20	31	401	1.92
AG/CT	-	618	291	138	65	32	43	1187	5.72
AT/AT	-	802	375	293	243	284	96	2093	10.08
CG/CG	-	7	2	1	-	-	-	10	0.05
AAC/GTT	294	98	71	3	-	-	1	467	2.25
AAG/CTT	815	296	138	9	-	-	5	1263	6.09
AAT/ATT	296	173	129	2	-	1	-	601	2.90
ACC/GGT	273	140	56	15	-	-	-	484	2.33
ACG/CGT	80	35	24	2	-	-	-	141	0.68
ACT/AGT	54	13	14	-	1	-	-	82	0.40
AGC/CTG	192	69	30	2	-	-	-	293	1.41
AGG/CCT	243	62	24	4	-	-	-	333	1.60
ATC/ATG	352	151	64	1	-	-	-	568	2.74
CCG/CGG	100	59	22	4	-	-	-	185	0.90
Others	162	29	6	1	-	1	2	201	0.97
Total	2861	2733	1328	529	342	5738	7225	20,756	
%	13.78	13.17	6.40	2.55	1.65	27.65	34.81		

## Data Availability

Not applicable.

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
