# Peer review of "Inflorescence Transcriptome Sequencing and Development of New EST-SSR Markers in Common Buckwheat (Fagopyrum esculentum)"

_plants, 2022, doi:10.3390/plants11060742_

Round 1
Reviewer 1 Report
I think the manuscript is recommended to be published in Plants but I just want authors to add more information about the differentially expressed genes as shown in Fig. 2. Authors just present the chlorophyll metabolism then you can add more data for the other traits. As you mentioned in the abstract, buckwheat has good nutrition, therefore, nutrient elements could be appropriate for fig. 3 or supplemental. I believe the information can attract more readers in Plants.
Author Response
Response: We gratefully appreciate for your valuable suggestion. Flavonoids are nutrients and anti-aging substances with high content in buckwheat. In Section 2.2, we added the differential expression analysis of key genes in the metabolic pathway of flavonoids, and attached Figure 3. The modification section uses "Track Changes" function and highlights for better viewing.

Reviewer 2 Report
Liu et al., performed RNA-seq analyses on two different buckwheat lines, in which they focused on the pedicels. Based on the sequenced results, they identified and developed EST-SSR markers that could be potentially employed for future genetic mapping or marker-assisted selection.
Major comment:
The reviewer found this study is very similar to Shi et al., 2017, which makes this study lacks novelty.
Other comments to the manuscript:
- The reviewer is curious why the authors chose the pedicles rather than floret or other tissues for RNA-seq?
- In section 2.1, the description of RNA sequencing results is rather ambiguous. Were the transcripts numbers shown in Table 1 representing the total reads from both Gr and UD? Did the authors remove the redundant reads? This should be clearly addressed.
- The EST-SSR markers designed in this study represent useful genetic information. Therefore, it is necessary to include the chromosomal locations of these markers, which were completely ignored by the authors.
- All legends to the figures or tables in this manuscript are not self-explanatory. More details should be added to improve the figure/table legends.
- References should be added for Fig.2B.
Author Response
Response: We gratefully thanks for the precious time the reviewer spent making constructive remarks. In response to these comments, we made the following reply, the modification section uses "Track Changes" function and highlights for better viewing.
- In fact, we selected varieties of common buckwheat with different pedicel thickness for RNA-seq rather than directly using pedicel sequencing, because there are two main differences between the green-flower buckwheat and the white-flower buckwheat, one is that the pedicel is thicker, and the other is that the petal color is green, and we want to find the difference genes in these two aspects at the same time, so the inflorescence is selected for RNA-seq.
- In section 2.1, we have enriched the RNA-seq results to highlight more detail. The raw reads after RNA-seq contains redundant sequences, and we got clean reads after removing redundant sequences and low-quality data. Subsequent analysis and the assembly of transcripts were also based on clean reads. The data in Table 1 are the specific data of the assembled transcript and Unigene.
- We have added the chromosome location of these primers, and please see Table S7 for specific data. Since there is no assembled chromosome data of the common buckwheat at present, we can only list the scoffold information.
- We added more detail to the figures and tables to show the data more clearly, such as Fig 2, Fig 3 and Fig 4.
- Figure.2B has added three references on chlorophyll metabolism, namely [34-36].

Reviewer 3 Report
The manuscript deals with an important topic today and has good potential. In its current state, however, it highlights numerous shortcomings.
Please find my comments below.
1. What gaps in knowledge would you like to address as the purpose of this paper? Presentation of technical research results may not be suitable to fill knowledge gaps in the field of your study. Please clearly address major challenges regarding the research topic and your novel contribution(s) to your field and in the context of your study.
2. I couldn't identify any hypotheses emphasized in the text.
3. I recommend adding some references to the latest subject literature (including Web of Science and Scopus papers).
4. I also believe the catalog of the literature cited is rather poor. I suggest expanding the list of literature studies by the years 2019-2021.
5. Please provide directions for further research and its practical implications
Author Response
Response: We gratefully appreciate for your valuable suggestion. In response to these comments, we have made the following changes, the specific parts of which are highlighted in the article in "Track Changes" function.
- At present, there are few studies on the molecular markers of the common buckwheat, which greatly limits the molecular breeding of buckwheat. In this study, we aim to develop a set of SSR markers of buckwheat, which will provide important help for the study of genetic diversity of buckwheat, the construction of genetic map and molecular assisted marker breeding.
- Thank you very much for your comments. As for the hypotheses in the article, please point out the specific hypotheses or paragraph so that we can modify or explain.
- We have added the latest references on chlorophyll metabolic pathways, namely [34-36], and we have also updated some references for RNA-seq, such as [52][58][60].
- For some references of a relatively old age, we have changed them to those of recent years, such as [5][10][15][20]. Unfortunately, we cannot change the references of some years to those of recent years, because there are few studies on buckwheat and literature updates are not frequent.
- In this study, after verifying the effectiveness and practicability of a set of SSR markers developed, we can use these markers to construct a genetic map of the common buckwheat, which is also of great significance to the collection and protection of buckwheat germplasm resources, and makes up for the shortage of common buckwheat SSR molecular markers.

Reviewer 4 Report
The manuscript entitled "Inflorescence transcriptome sequencing and development of new EST-SSR markers in common buckwheat (Fagopyrum esculentum)" is an original work of great scientific value. The plant material studied by the authors was Fagopyron esculentum Moench., an edible crop of recognized medicinal value that easily adopts to various edaphic and climatic characteristics. In their study Illumina HiSeq 4000 hi-throughput sequencing technology was chosen to obtain RNA from two kind of materials, that is green-flower common buckwheat (Gr_F) and white Ukrainian daliqiao (UD_F). Briefly, they obtained a total of 118 448 unigenes with a total sequence length of 147 868,721 bp, and 77 424 unigenes were annotated in at least one database. A total of 20 56 EST-SSR loci were mined in 17 595 unigenes, and they developed 13 909 pairs of primers. Furthermore, the authors, after preliminary screening, selected 224 pairs of primers were, and then homology clustering analysis were done with 35 varieties of common buckwheat and 13 varieties of Tartary buckwheat The 48 buckwheat varieties were divided into two groups according to the similarity coefficient of 0.68. Authors concluded that the results of transcriptome sequencing and assembly, primer sequencing and differential expression analysis may provide a theoretical basis for species classification, germplasm conservation, genetic diversity analysis and above all molecular marker-assisted breeding of buckwheat.
Some comments on the possible improvement of the text:
- From abstract, please delete the following phrases: line 11 - (1) Background, line13: - (2) Methods, line 16 - (3) Results, lines 24/25 - (4) Conclusion
- Introduction, Line 30 - Fagopyrum esculentum M. It should be: Fagopyron esculentum Moench.
- Results, Fig. 1 - Please, do improve the readability of the description of the graphs.
- References - It is recommended to adapt the record of the publication to the journal's requirements. In addition, pay special attention to the completely incorrect form on lines 516 and 517.
These are just a few of the very small improvements that I propose. The other elements of the work do not raise my doubts.
Author Response
Response: We are very sorry for our incorrect writing, and we have made correction according to the Reviewer’s comments as follows, the modification section uses "Track Changes" function and highlights for better viewing.
- In the abstract section, we have removed a few words that you pointed out.
- We have made changes as requested.
- We have described Figure 1 in more detail in section 2.1 and added new references, such as [31-33].
- We have made systematic changes to the format of the references.

Round 2
Reviewer 2 Report
The authors addressed most of the comments. However, the whole manuscript lacks proper hypotheses or research questions, which was also raised by another reviewer.
In the responses to my first comment, the authors mentioned the difference between green-flower and white-flower pedicels, which constitute one important question in this research. If this piece of information can be added, it will largely improve the whole manuscript.
Author Response
Response:We would like to thank you for your careful reading, helpful comments, and constructive suggestions, which has significantly improved the presentation of our manuscript. The modification section uses "Track Changes" function and highlights for better viewing.
In Section 2.2, we also elaborated the differentially expressed genes in flavonoid and chlorophyll synthesis pathways. In the discussion section, we conducted differential gene analysis based on the transcriptomic data of Gr and UD. In the chlorophyll pathway, we found 33 differentially expressed genes between them, among which 27 genes were up-regulated in Gr. We speculated that the green-flower buckwheat is mainly caused by chlorophyll related genes. The specific paragraphs are in lines 156-185 and 259-271.
In addition, we also added pictures of traits difference analysis of Gr and UD, including electron microscopy of pedicel thickness, chlorophyll contents and column chart of pedicel thickness, please see the attachment.

Round 3
Reviewer 2 Report
The authors addressed my comments. Now I think this manuscript is suitable for publishing in Plants.